# Complexity in Biological Organization: Deconstruction (and Subsequent Restating) of Key Concepts

**DOI:** 10.3390/e22080885

**Published:** 2020-08-12

**Authors:** Mariano Bizzarri, Oleg Naimark, José Nieto-Villar, Valeria Fedeli, Alessandro Giuliani

**Affiliations:** 1Department of Experimental Medicine, University La Sapienza, via Scarpa 16, 00160 Rome, Italy; valeriafedeli9@gmail.com; 2Via Scarpa 16, SBGLab Group, University Sapienza, 00160 Rome, Italy; 3Institute of Continuous Media Mechanics UB RAS, Perm 614013, Russia; naimark@icmm.ru; 4Department of Chemical-Physics, A. Alzola Group of Thermodynamics of Complex Systems of M.V. Lomonosov Chair, Faculty of Chemistry, University of Havana, Havana 10400, Cuba; nieto@fq.uh.cu; 5National Health Institute, 00161 Rome, Italy; alessandro.giuliani@iss.it

**Keywords:** complexity, nonlinearity, system, symmetry breaking, emergence

## Abstract

The “magic” word complexity evokes a multitude of meanings that obscure its real sense. Here we try and generate a bottom-up reconstruction of the deep sense of complexity by looking at the convergence of different features shared by complex systems. We specifically focus on complexity in biology but stressing the similarities with analogous features encountered in inanimate and artefactual systems in order to track an integrative path toward a new “mainstream” of science overcoming the actual fragmentation of scientific culture.

## 1. Complexity: Deconstruction of a Word

What we do mean by “biological complexity”? A number of attempts have been made to catch the meaning and the implications that such a word actually has to lay out the fundamentals of a theory of biological organization. It is bewildering that, in most cases, we must resign ourselves to self-evident statements like the following: “In a general sense, the adjective “complex” describes a system or component that by design or function or both is difficult to understand and verify [1]. Yet, we are beginning to realize that both natural as well as artefactual complex systems share many universal design patterns and are built on the same principles. Identifying such principles will enable grasping the rules and the constraints that govern complex systems functioning. In turn, phenomenological modeling of such systems can help in generating and testing hypotheses on such principles [2]. Complex systems modeling encompasses a different attitude with respect to classical differential equation style adopted for situations in which we can rely on a dee knowledge of basic theoretical principles because of the lack of knowledge of such principles [3]. Up to now, we can confidently recognize the overwhelming importance of the context in constraining the way the general laws of nature that actively “shape” the specific complexity under scrutiny. In the following sections, we will discuss and clarify this point further.

We think that a fair way to face with this challenge implies to begin by “deconstructing” the word and exploring its etymological meaning. Complex means “composed of interconnected parts, formed by a combination of simple things or elements.” The lemma “complex” comes from Latin complexus “surrounding, encompassing,” past participle of complecti, “to encircle, embrace,” from -com “with, together” + plectere, “to weave, braid, twine, entwine” [4]. First appearance of the word can be traced back to 1650, while its routine use is attested since 1776, when development of classical physics just met some intractable problem associated with complex phenomenon, like those posed by hydrodynamics and heat flow [5]. Later on, complexity emerged in biology from the experiment of Hans Driesch showing how a whole organism could be produced from single cells harvested during the first duplication cycles of the zygote, before the developing embryo reach a full maturation state. Driesch studied sea urchin embryos and found that when he separated the two cells of the embryo after the first cell division, each developed into a complete sea urchin [6]. Nonetheless, it was not until the second half of the 20th century that it became evident that science has succeeded in solving a bewildering number of relatively easy problems, “whereas the hard problems and the ones which perhaps promise most for man’s future, lie ahead” [7].

In his seminal paper, Warren Weaver stressed that we are usually facing three order of situations about complexity: (1) organized simplicity, where the relationships among particles do not change across time and laws of classical physics holds up; (2) disorganized complexity, where the behavior of a huge number of entities—each of one has an individually erratic or unknowable behavior—can be caught by looking at macroscopic parameters governed by laws of statistical mechanics (thermodynamics), altogether with the support of probability theory; (3) organized complexity, a specific feature of systems (like the living ones) where several entities (besides being not enough to rely on a “simple” statistical mechanics approach) establish relationships that can change over time, enabling the overall system in acquiring a form, according to organizing principles. This latter kind of complexity occupies a pivotal place at the mesoscopic scale—an intermediate position between atomic and macroscopic dimensions—where entities interact so allowing coherent “organization” to emerge. This kind of complexity allows collective organizing principle to “arise” from microscopic rules, while being—strictly speaking—independent of them [8]. Both biology and medicine deal with problems posed by organized complexity for which, to date, a compelling scientific discipline is still warranted. However, we can now identify the mesoscopic realm the level where objects organize themselves and function in ways unlike anything we know at very large or very small scales, thus displaying astonishing and unpredictable properties, as seen in both living and nonliving systems where apparently spontaneous organization processes take place.

## 2. Systems

In the above definitions, another word comes into play: system. The lemma comes from Greek through Latin—i.e., “organized whole,” a whole compounded of parts—from the stem of the verb *synistanai* “to place together, organize, form in order,” from *syn-* “together.” Overall, system means a “set of correlated principles, facts, and ideas.” Again, as for the word “complex,” “system” appears at the beginning of the 17th century and was then applied to identify, animal body as an “organized whole” since 1680 [9]. Here, we recognize a system as an “integrated whole composed of different interacting parts” that fall under observation. This means that we must consider a coherent whole, provided of autonomous behavior, in order to investigate its structure and intrinsic dynamics.

To do so, we have first to choose a proper level of observation, specifically the one where the mesoscopic properties arise. Understanding the logic of organisms implies the possibility to recognize correlations between the *local* processes and the *global* structure of the living beings, connecting different organization layers. The existence of between layers correlations stems from the fact that molecules, molecular aggregates, organelles, cells, tissues, and organs are constrained to cooperate to keep the functionality of the whole. These constraints lie in the boundary and initial conditions, so that “the organization becomes cause in the matter” [10]. In the following, we will go in depth into the identification of the proper level of observation where to look in order to maximize the probability of catching such across levels correlations.

Second, how the system can be described, i.e., what are the variables (the “observables” in the language of physics) to look at? In order to give some hints on the choice of variables to consider, it is worth noting that any system is delineated by spatial and temporal boundaries, through which the system dynamically interact with the environment.

This makes interface variables as the most promising field of investigation. It is worth stressing that some sort of organized interfaces emerge anytime when a complex system must face the (largely unpredictable) solicitations coming from its environment. The structure and function of such an interface is then a privileged observatory on the system itself.

Interfaces provide a system with a recognizable structure, as such epitomized by a specific *form*, tightly committed in performing well-defined tasks and functions. This prompts to look at size invariant relations among descriptors that encompass the concept of *shape*.

Lastly, a complex, living systems evolves over time through successive critical transitions, reaching different stable-states, which are usually depicted altogether by a Waddington-like landscape [11]. This suggests that we must privilege shape and dynamical descriptors that can be observed along relevant time courses over static variables.

## 3. Hierarchy of Levels and Emergence

Emergence, a concept first introduced by Aristotle [12], was reintroduced in the scientific debate by G.H. Lewes in the 19th century to indicate, “Something new which could not be predicted from the elements constituting the preceding condition” [13]. Noticeably, these features “emerge” only when parts interact in a wider (“organized”) “whole.” It is noticeably that the word (from Latin ex “out” + *mergere* “to dip, sink”) evokes the rising from a liquid by virtue of buoyancy, thus suggesting how the interaction among upward forces at the lower levels can produce recognizable effects at a level higher than that in which interactions among components take place. Therefore, understanding those properties require that the description of complex processes ought to be performed by looking at the level at which these properties “emerge.” An emergent property need not be more complicated than the underlying non-emergent properties that generate it. For instance, the laws of thermodynamics are remarkably simple, even if the laws that govern the interactions between component particles are usually more “complicated” and frequently mathematically intractable.

Emergence has been reported in purely physical settings as well as in living organisms. An example from physics of such emergence is water, which behavior appears unpredictable even after an exhaustive study of the properties of its constituent atoms [14]. Another well-known example of emergent properties from inanimate matter is superconductivity. Superconductivity is the set of physical properties observed in certain materials, wherein electrical resistance vanishes and from which magnetic flux fields are expelled. Any material exhibiting these properties is a superconductor. Unlike an ordinary metallic conductor—whose resistance decreases gradually as its temperature is lowered—a superconductor has a characteristic critical temperature below which the resistance drops abruptly to zero. Therefore, an electric current through a loop of superconducting wire can persist indefinitely with no power source (so creating a sort of perpetual motion). Superconductivity, as well as superfluidity, are collective phenomena owing their existence to many-body interactions occurring under a specific regimen of constraints (like temperature). The corresponding emergent states are not related perturbatively to the parent state. Therefore, in understanding such properties, we cannot focus on microscopic properties of the normal state, as the transition from the microscopic parent state to the collective emergent state is not analytic [15]. To quote Anderson’s words, “the whole is not only greater than but very different from the sum of the parts” [16]. Thus, we are asked to focus mostly on (1) relationships and (2) system’s constraints, both internal and external. The dynamical interplay between these two factors is not only required to allow emergence to occur but can also explain how, during that process, the system undergoes a critical transition towards a higher level or organization. Thereby, a hierarchy of levels is needed to obtain a compelling description of the system [17]. Consequently, the widely used concept of “physical (scientific) law”—as far as it has been borrowed from jurisprudence—has become inappropriate. Inferring causality from a law is an undue logical conclusion, a critical issue discussed in science since the time of Hume. As relationships changes with the level of organization of the system, so do the causative factor(s). Causal relationships (as discussed later) may display scale-dependence, while scientific laws regulating the phenomena observed at lower levels, could be inadequate to accommodate with processes recorded at higher levels. Indeed, scientific laws are intended to be laden neither with ontological commitments nor with statements of logical absolutes. A law means here a set of relationships, deemed to reflect—even if it is not explicitly asserted most of the time—a causal relationship between the observed phenomena [18]. For these relationships changes according to the level of observation we are actually considering, the regulatory power of the law does not hold at every level. Indeed, some (scientific) laws reflect mathematical symmetries found in nature, and, as such, can be broken when the system experiences a symmetry breaking. In other worlds: distinct rules are required in explaining phenomena emerging at different levels.

This statement introduces some very relevant questions given that emergent phenomena share some common—universal traits that are largely insensitive to changes in their microphysical base, as pointed out by studies of the Renormalization Group [19]. “Universality” refers to the fact that phase transitions arising in different systems often possess the same set of critical exponents, while the thermodynamic properties of a system near a phase transition depend only on a small number of features, such as dimensionality and symmetry, and are insensitive to the underlying microphysics. Conclusively, *emergence is not an epistemic construct*. Instead, it reflects a true ontological reality shared by complex systems of very different nature [20].

## 4. Symmetry Breaking

Transitions between different levels of organization, associated with appearance of emergent properties, call into question a pivotal aspect of complex systems: symmetry breaking.

By symmetry, we mean the existence of different viewpoints from which the system appears the same (superimposition). It is worth noting that the symmetry present in the microscopic equations is not present in the macroscopic system, due to phase transitions. According to Laughlin, while for many particle systems, nothing can be calculated exactly from the microscopic equations, mesoscopic and macroscopic levels can be effectively described in their own terminology and have properties that do not depend on many microscopic details. This does not mean that the microscopic interactions are irrelevant, but simply that you do not see them anymore—you only see a renormalized effect of them [21].

Let us to consider water across its phase transition as an example. When water is in a liquid state there is no privileged direction (the system is isotropic, that is to say “symmetric by rotations”), whereas when water is solidified as ice, it assumes a crystalline structure with spatially periodic patterns. This implies that the system is no longer symmetric by continuous rotations: it has a few privileged directions determined by its crystalline structure, which means that it has a smaller symmetry group (less spatial symmetries). These conditions are mirrored by parallel changes in the system’s energy balance: in the liquid state, entropy dominates, associated with a more disordered phase, while in the crystalline, more ordered phase, energy dominates. In a broad sense, a system that travel across different symmetry breakings to reach configuration characterized by low symmetry moves from a disordered to a more ordered condition. Living objects, given that they are placed in between the full disordered systems (gas) and the full ordered solids (crystal), show an intermediate value of order, subject to a regimen of extended critical transitions, i.e., a succession of symmetry breaking events [22].

As a consequence, symmetry-based explanations using symmetry breaking (SB) as the key explanatory tool have complemented and replaced traditional causal explanations in various domains of physics [23]. The development of our Universe, as well as the appearance of life on Earth, have been ascribed to a sequence of symmetry breaking events. Similarly, cell differentiation since the first stages of embryo development—when each cell retains totipotency and the system is totally homogeneous and symmetric—begins with a symmetry breaking, epitomized by gastrulation, mostly triggered by a physical external force deployed by contact of the blastocyst at the bottom level with the myometrium. By sensing this force, the single layer of cells arranged in the cylindrical egg is ultimately exposed to an asymmetrical stimulus that compresses the inner portions of the cells, pushing them inwards and creating a bulge. Soon, the cell layer folds within itself and starts to differentiate [24]. Across this succession of symmetry breaking events, the system rearranges the relationships among its internal components as well as with the environment, thus assuming different configurations, which, in a living organism, lead to the emergence of an organized whole recognizable by its form.

## 5. Form and Relationships

Downstream a symmetry breaking, the way the entities are connected among themselves dictates the shape—the form—the system will assume. In other words, the form epitomizes the entirety of relationships the components establish among them. In turn, the form influences the dynamical relationships among components as well as with the external milieu [25]. According to recent reappraisal of Aristotle’s work, form is what unifies different material components (matter as substance) into a single object. This statement highlights a number of key elements: the number of components that are involved, the type of relationships they establish, and how these interactions are intertwined and compartmentalized. Indeed, especially in living cells, topological segregation (i.e., confinement within cytosol membrane, on organelles, upon protein-based scaffolds and upon the cytoskeleton) plays a critical role in governing metabolic and functional processes [26]. Overall, the shape allows the canalization of several processes: the structure literally bends the interacting components and their dynamics, leading to dramatic increases in the efficiency of signal transfer as well as to enhanced specificity of signal flow, despite possible cross-reactivity with other pathways that are apparent in the test tube [27]. Any modification of the system’s architecture can canalize the biological processes in a very different way. This is a completely different picture with respect to what happens in simplified systems or in test tube. In these latter conditions, reactions among a few components are governed by simple dynamical rules and the output of the reaction is completely determined by the concentrations of each of the components and the reaction rates. Instead, within a complex structured system, chemical processes and biological functions are shaped by the three-dimensional context in which they occur. In fact, an impressive body of evidence demonstrated that by physically modifying the shape of a cell, we can trigger a bewildering number of different outputs: from differentiation to apoptosis, including even reversion of a cancerous phenotype [28,29,30]. This means that an unexpected level of causality emerges uniquely from the specific configuration adopted by the system.

Overall, these investigations provided further support to the issue raised by Folkman and Moscona in 1970s [31]. Relevance of the three-dimensional configuration becomes evident at different scales of living organism organization, from organs to complex molecular networks starting from protein contact networks (PCN) [32] to gene expression and metabolic networks. Indeed, the specific form displayed by a molecular network (correspondent to its wiring architecture) is more relevant in fulfilling specific functions, than the analytical description of the constitutive microscopic components. In fact, tightly regulated network behavior can result from widely disparate molecular values, suggesting that even considerable variability in many parameters do not hinder the network function, as long as the overall network configuration (its shape) is preserved [33,34]. Mikulecky clarifies this issue in terms of “network thermodynamics” [35].

Making a long story short, any system is governed by both constitutive and network laws. Constitutive laws refer to the physics governing single elements (e.g., capacitors or resistors in an electrical circuit), whereas network laws stem from the specific wiring architecture irrespective of the physical nature of its elements, i.e., a social network has emergent properties identical to an electrical circuit if they share the same wiring architecture. Thus, we can sketch a raw definition of mesoscopic realm as the place where network laws prevail. This opens the way to a new quest for universal principles based on the assumption that any system is made of interacting parts. Therefore, it is not by accident that shape displays a relevant functional and heuristic (diagnostic) role in several biological and medical settings [36], especially when quantitative shape changes are associated to phenotype transition, as such those occurring in carcinogenesis or in tumor reversion [37,38].

The same topology, which gives rise to the same network properties, can result from infinite collections of different elements. Therefore, there is a level of causality and explanation that is purely relational in nature and that does not refer to the objects involved in it [39]. This means that in organized complex systems, the construction of the networks is neither casual nor arbitrary. Topological constraints impose restrictions and determine preferential topologies given that the connections are hierarchically organized, i.e., they obey to a system logic, largely independent from the underlying microscopic activity. The dynamical relationships that concur in establishing the system’s shape possess some critical features that deserve a detailed survey. Namely, scale-dependence, nonlinearity, and space-time evolution of networks show to be critical for system’s description.

## 6. Scale Dependence

The literal meaning of scale is the unit by which we can quantify an observed quantity. We can infer the scale (in terms of measurement unit) from the limitation of the proportionality relation. We reach the scale that determines “smallness” when proportionality relations are destroyed. In other words, the scale problem emphasizes that “levels of organization can be thought of as local maxima of regularity and predictability in the phase space of alternative modes of organization of matter” [40]. Therefore, in choosing a proper scale, we have to look where pattern and regularities are at a maximum. This request has an immediate statistical counterpart in the maximization of nontrivial determinism [41], which translates into the operational suggestion to put your analysis at the level that maximizes the number of empirical correlations. These statements carry a huge number of critical questions. First, as organization levels are discrete and scales are continuous quantities, we have to identify a scale range along which the relationships among entities display robustness and invariance (organs, tissues, cells, organelles, molecules, and atoms). Second, processes can be appreciated—and mandatory be studied—at those levels of organization at which they actually happen [42] and where we may find peaks of regularity or predictability or clusters of causal relationships [43]. Third, the entrenched interactions among different organizing levels may help in shedding light to the controversial issue of causation in biology. Given that higher level of organization can interact and even superimpose their driving force upon lower levels, it has been hypothesized that the behavior of lower level things is constrained by the higher level whole that they are a part of. Accordingly, Campbell expressed this idea as follows: “processes at the lower levels of a hierarchy are restrained by and act in conformity to the laws of the higher level” [44].

It is worth noting that the reductionist approach—as well as linear theory—keeps silent about the scale—it is “unaware” of the scale, and in this way misses the complexity of living objects [45]. Addressing the scale problem entails the deconstruction of the linear theory, focusing attention on the world of large scales where the proportionality relation distorts and linear theory ceases to apply.

## 7. Nonlinearity

Relationships in complex systems are mostly nonlinear. A linear system is governed by linear differential equations, and it obeys the law of superposition, thereby ensuring that outputs will be proportional to inputs. On the contrary, nonlinear systems are governed by a set of simultaneous (nonlinear) equations in which the unknowns (or the unknown functions in the case of differential equations) appear as variables of a polynomial of degree higher than one or in the argument of a function which is not a polynomial of degree one. In these conditions, changes in variables over time may appear chaotic, unpredictable, or counterintuitive, contrasting with much simpler linear systems. In other words, nonlinear dynamics is at the root of emergent phenomena as this kind of output is precisely the consequence of nonlinearity. In addition, nonlinear systems display the astonishing property of extreme sensitivity to the initial conditions, which further add unpredictability and lay out the way for the appearance of true novelty. Despite physical sciences have been “revolutionized” by the (re)discovery of nonlinearity since 1970s, biologists have been reluctant in accepting it. Many biologists still turn to reductive explanations based on well-defined causal chains, thereby missing the implications of emergence for understanding their phenomena. Sad to say, first attempts to introduce nonlinear thought into chemistry and biology have been disappointing, leading to a substantial rejection by the mainstream, the seminal contributions made by Herbert Fröhlich notwithstanding [46]. Vicissitudes experienced by scientists, like Belousov (whose findings were rejected by the Soviet scientific establishment), Colin McClare, or Giorgio Careri, demonstrated that the scientific suspicion or rejection of nonlinear theory in biological sciences was a worldwide phenomenon until recently [47].

Taking seriously, nonlinearity implies a critical appraisal of ergodicity. Ergodic systems are defined as those for which the time spent by the systems in a region of the phase space of microstates is proportional to the size of that region. In other words, in the long term, all accessible microstates are equiprobable and are all eventually visited. In contrast, non-ergodic systems do not visit all microstates and possess memory of the initial state. They are characterized by historical memory whereby the present state of the system depends on the previous history, and accordingly, the future states are not precisely predictable. This non-ergodic behavior is caused by competing short- and long-range interactions, which result in frustration and can produce complex patterns, including bistability and hysteresis [48].

Processes of this type can be described by a space-phase diagram, first introduced by Poincaré and then applied by Prigogine to the study of irreversible phenomena [49]. According to the theory of dynamic systems, the space of the phases of a system is a space with each possible state corresponding to one unique point in the phase space. In a general sense, the phase space describes the trajectories of a system that reach areas of high stability (attractors), by travelling across bifurcation points and unstable regions. Attractors and unstable regions are usually depicted as valleys and hill in the Waddington’s metaphor, which depict the phase space as a landscape [50].

Classically, a number of differential ordinary equations (ODE) can be used to describe a dynamical system, which is recognizable in the phase space by a vector, obtained from system’s physical quantities (variables or observables). Linear equations of the (classical) thermodynamics of reversible processes (belonging to closed systems in opposition to open systems like those occurring in living organisms) are usually analyzed in terms of fixed-point attractors. The system is “attracted” to the steady state of minimal energy and tends to “forget” its initial conditions. On the other hand, nonlinear systems have more than one solution: new, unexpected “solutions” emerge when the system passes through a bifurcation point, eventually going towards a catastrophe [51]. The system’s behavior at the bifurcation point depends on the previous history and thus, the initial conditions cannot longer be forgotten. The Belousov–Zhabotinsky (BZ) reaction provides an excellent example of how, even a reversible process, when maintained in a non-equilibrium steady state, can promote the emergence of nonlinearities [52]. If the concentrations of the different chemical species in the BZ reaction are kept constant (by removing or adding one or more components or shaking the solution), the equilibrium cannot be reached. This condition is known as “stationary state of non-equilibrium.” Being potentially unstable, it shall give rise to unpredictable fluctuations, which ultimately foster organizing phenomena, governed by long-term correlations. In this setting, we may observe that ODEs lead to the appearance of a strange attractor: the interaction between stationary dissipative structures and propagating waves enables the emergence of self-organized forms [53]. Similarly, since the seminal study published by E.N. Lorenz—regarding the long-term unpredictability of weather forecast [54]—a number of reports have highlighted how complex nonlinear systems—both in inanimate matter and in living organisms—can give rise to self-organizing processes, showing a bewildering dependence in respect to even mild fluctuations of a few initial conditions. This is especially relevant in biology, where a huge number of parameters can significantly modulate a biological process towards very different outcomes in an utterly unpredictable way [55,56]. Yet, due to the strong relationships among a stable attractor (recognized by a high order of connectivity among the single components), a complex system can display at the same time a significant robustness, i.e., resilience in respect to external/internal perturbations [57].

To sum up, complex systems, governed by nonlinear dynamics undergo a sequence of modifications across space and time, through subsequent symmetry breakings downstream to bifurcation points. This behavior allows systems to reach different attractors (i.e., stable states identified by specific boundaries and distinctive shapes) by travelling across a number of critical transitions.

## 8. Critical Transitions

This approach highlights how relevant are symmetry breaking and critical transitions in allowing a system to assume new configurations, ultimately producing true novelty in biological systems. This model has been applied especially in understanding biological morphogenetic models. Living cells, starting from a complete symmetric configuration in which the genome express totipotency, are committed towards cell fate specification by which cells differentiate, acquiring different (quasi) stable phenotypes (i.e., attractors). Along this process, cells reorganize their genome pattern of expression by rewiring the gene regulatory network as well as their shape, involving in such a complex activity of thousands of genes, proteins, cytoskeletal components, and many other intracellular structures. The coordinated control of a huge number of entities must overcome several hurdles (intrinsic instability in gene expression due to stochastic fluctuations and thermodynamics barriers) and integrate a disparate set of external and structural constraints that fold disorganized cell processes into a specific (deterministic) outcome. It is amazing that such a “Brownian-like” turbulence—reflecting the disordered behavior of the microscopic level might be harnessed through only a small number of control parameters that underpin highly complex morphogenetic processes [58]. This approach, remnant of the conceptualization established for understanding the phase transition occurring in inanimate gas–liquid systems, has been conceptualized according to the self-organized criticality (SOC) [59] or by other models based on downward causation in which internal and external driving forces are integrated into a whole system [60]. Indeed, it has been observed—both at the population and single-cell levels—that a sandpile-type critical transition self-organizes the overall molecular “trafficking” into a few transcription response domains (critical states). A cell-fate change occurs by means of a dissipative pulse-like global perturbation in self-organization through the erasure of initial-state critical behaviors (criticality). It is noticeable that rewiring of genome expression associated with the emergence of a new self-organized phenotype involve a global destabilization of the previous state when approaching a bifurcation point. This phenomenological approach does not require detailed modelling of the dynamics of the underlying gene regulatory network (GRN), nor the identification of a bifurcation parameter, which is currently not realistic given our insufficient knowledge of the GRN architecture. The key formal assumption is only that the GRN state change implementing the cell fate decision and commitment is due to a monotonical gradual alteration of the value of an (unidentified) bifurcation parameter that drives the change of the attractor landscape through a bifurcation (without specifying which type) [61]. However, this model can hardly explain how cell can choose among different cell fates if external cues and driving constraints are left unconsidered.

Indeed, the genotype-to-phenotype mapping is not one-to-one but one-to-many [62], given that as the number of regulatory interactions grows, so does the number of phenotypes that can potentially be sampled [63]. As a fact, strictly deterministic models of cell differentiation are contradicted by experimental evidence, highlighting that gene expression is a stochastic process. Instead, constraints can contribute to determine a directionality to the differentiating commitment, as vindicated by a compelling body of evidence [64].

Constraints belong to two classes: (a) holonomic (independent of the system’s dynamical states, as being established by the space-time geometry of the field) and, (b) non-holonomic (modified during those biological processes to which they contribute in shaping) [65].

This latter kind of “constraints,” in which dynamics works on the constraint to recreate them, have emerged as critical determinants of self-organizing systems, by manifesting a “closure of constraints” [66]. Overall, the constraints act by harnessing the “randomness” represented by the simultaneous presence of equiprobable events restraining the system within one attractor. Furthermore, coordinated constraints can work their way around physical laws, as emphasized by studies, which demonstrated that, when introduced in a field constrained by specific colloid structures, like-charges can surprisingly attract themselves, instead to be repulsed, as posited by Coulomb force [67,68]. This example epitomized a further aspect of constrained self-organizing systems, i.e., the capability to achieve a distinct level of causation, operating in addition to physical laws, generated by the action of material structures acting as constraints.

A comprehensive model, based on specific organizing, biological principles, has highlighted how, according to such premises, self-organizing processes, constrained by both physical and auto-generated constraints are instrumental in driving morphogenetic processes, given that “evolution” cannot be explained only in terms of natural selection [69].

Biological processes involve assembled molecular components (open complexes) responsible for mesoscopic properties, spatial-temporal multiscale organization, according to the epigenetic (thermodynamic) landscape in which system’s transitions occur [70]. Thermodynamic aspects of the open complexes are of utmost importance. According to this frame, “structural defects” (such as local distortions of the global structure, e.g., breaks along DNA double helix) play a crucial role. Defects give rise to “distortion modes” invading the entire system that allowed the formulation of thermodynamics of the open complexes and the evolution equation in the generalized Ginzburg–Landau form reflecting specific type of criticality named as the structural-scaling transition [71].

These distortion modes have three basic forms we can define as “breathers” “solitary waves or solitons,” and “blow-up” modes. The first form is responsible for the oscillation of the system around its equilibrium position; the name “breather” derives from the continuous local opening/closing of DNA double helix making transcription possible caused by localized and transient interruptions (nicks) of the helix. This is “business-as-usual” and corresponds to the continuous “small-avalanches” happening in a sandpile that do not alter its slope (subcritical state in SOC). The solitons [72] are solitary waves traveling the entire structure corresponding to a critical state generating long-range correlations. The most dramatic “blow-up” behavior implies a disruption of the structure that literally explodes for a sort of domino effect. Ginzburg and Landau were able to set the values of the critical points correspondent to the progressive transition between these subsequent attractors by means of a structural parameter d (with d1 < d2 < d3 as corresponding transition points) typical of each considered system [73].

This result is of utmost importance in biological complexity given that it establishes a deep structure/function link we can directly observe by experimental methods [74].

## 9. Concluding Remarks

In the search for a definition of the word “complex,” we encountered a number of relevant issues that can help in providing a compelling set of identification rules. Physics always tries to find a precise and rigorous definition of its concepts, so the question: What is meant by complexity? Physicist Seth Lloyd [75] collected 31 definitions of this concept, divided into groups according to its genesis. Currently, this list includes more than 45 definitions, among which are: Shannon entropy, Gibbs-Boltzmann entropy, algorithmic complexity (Chaitin, Solomonoff, and Kolmogorov), Renyi entropy, Tsallis entropy, Kolmogorov entropy, fractal dimension, among many others. From the complexity exhibited by dynamical systems, we must highlight the following general aspects essential to understand this phenomenon: (1) complex, it should not be seen as a synonym of complicated, since a system described by few degrees of freedom can exhibit a high complexity during its evolution; (2) complexity leads to the appearance of emergent properties [76], with their corresponding macroscopic observables; and (3) complex systems display temporal and spatial patterns exceeding their topological dimension that cannot be reduced to an integer and, consequently, require to be expressed by a fractal dimension [77]. (4) Complex systems display a nonlinear dynamics that frequently shows a sensitive dependence to the initial conditions, a behavior that can be confused with stochasticity [78]. The most important consequence of this property is the impossibility of making predictions about long-term evolution. In other words, the so-called Laplacian determinism collapses. (5) Finally, for a deterministic dynamic system to exhibit complex behavior, two fundamental requirements should be fulfilled: nonlinearity and existence of feedback processes.

Noteworthy, the fundamental mechanism that describes the emergent properties and complexity of a system is based on the occurrence of bifurcations that enact the occurrence of symmetry breakings [79]. Bifurcations in dynamical systems are analogous to phase transitions in the proximity of equilibrium [80] and result from the amplification of microscopic fluctuations at the macroscopic level. Ultimately, this process leads to the emergence of new configurations, i.e., new pattern of (self)-organization of the system within a different attractor in the Waddington’s landscape [81,82]. A critical role in displacing a system from its previous stable state, or alternatively in stabilizing its current basin of attraction, is provided by external perturbations. Dynamic systems may or may not be controlled by the effect of periodic external fluctuations, according to the type of dynamic systems complexity [83]. Sensitivity of the system to external fluctuations depends on its robustness, i.e., on the capability to maintain its functionality against various external and internal challenges, including pharmacological stresses [84]. It is noteworthy that the robustness of tumor cells is a consequence of the phenotypic and genomic heterogeneity of cancer cells, which ultimately contribute in shaping the fractal geometry of cancerous cells and tissues [85]. Overall, the development of a primary tumor from a microscopic level (avascular growth) to a macroscopic level (vascular phase), and the subsequent appearance of metastases, is not simply the accumulation of malignant cells, but results from a nonlinear process involving true “biological phase transition” downstream critical bifurcations [86,87]. This dynamical behavior leads to self-organization away from thermodynamic equilibrium, providing the system with a high degree of robustness, complexity, and hierarchy [88], which, in turn, enacts the creation of new information and learning ability.

Although there may be many nominal degrees of freedom available, the system may organize its motion through the Waddington’s like landscape into only a few effective degrees of freedom than those nominally accessible. The system is attracted toward a lower dimensional phase-space, favoring the development of specific, recognizable structures (the “forms”) provided by self-organizing processes. The interplay among internal/external cues, coupled with the presence of constraints, which act through a sequence of extended criticalities, reduces progressively the degrees of freedom of the system, until an outcome is reached [89]. A similar trend can be observed during the shift from one morphotype to another in the course of the specialization/differentiation of the cell lineage: a cell type proceeds through a discrete number of morphotypes along its specializing/differentiating pathway and every morphotype could be considered as a stable steady state [90].

The most important lesson we can derive from the above-sketched “Complexity Portrait” outlines the relevance of too often neglected *details*—including the scale, the correlation in between structure changes, transitions over time of the pattern organization, and fluctuations in system’s robustness—in shaping the overall system’s function. However, the experimental approach needs to translate from the continuous multivariate description of the system into discrete configurations: such discrete configurations are the hidden treasure we must look for in our scientific enterprises. A schematic picture regarding how levels are intertwined across different scales is described in Figure 1.

Within this landscape, complexity degrees, occurrence of constraints, and entropy fluctuations proceed in opposite way, ultimately giving rise to an ordered structure while starting from a chaotic/stochastic microscopic condition.

We want to spend a few words about the methodological tools by which we can put the advocated complexity definition at work. It is normal for a physicist to associate the study of dynamics to a set of differential equations but this approach is practically unfeasible in biological systems that are nonstationary and lack of a sufficient number of experimental points to avoid chance correlation and model indeterminacy. The same kind of information (even if at a different detail) can be obtained by computational unsupervised approaches like principal component analysis [91,92] or Recurrence Quantification Analysis [93].

On the same heading, a correlation matrix has the same mathematical structure of a graph adjacency matrix that in turn has a one-to-one correspondence with the wiring of a complex network. That is to say that the mathematics of complexity has “no clothes” and that the methodological choices only stem from the contingent features of the studied phenomena. This methodological freedom stems from the shared organized complexity character (with the consequent emphasis on relational structure) of systems coming from all the science fields. This shared character is exactly what we tried to describe in this essay.

## Figures and Tables

**Figure 1 entropy-22-00885-f001:**
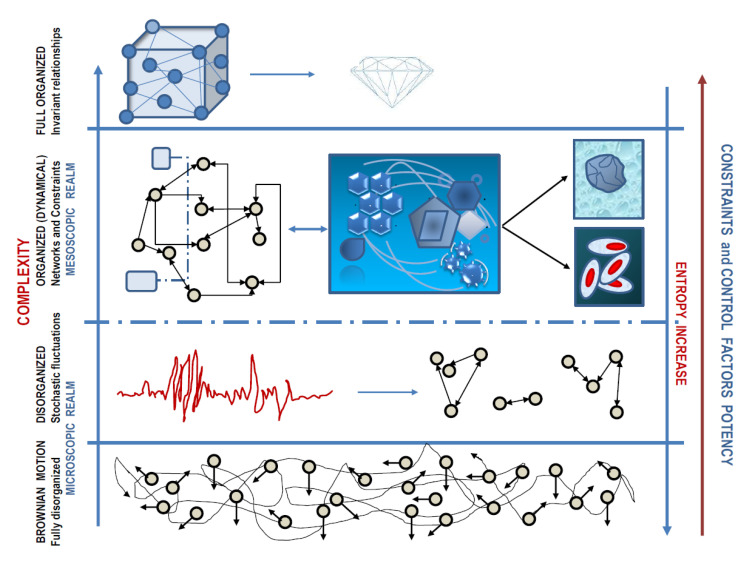
*Evolution of complexity: degree and architecture*. At the lowest level (atoms and simple molecules), behavior is mostly chaotic (Brownian motion) and essentially governed by thermodynamic boundaries; constraints are almost absent. The number of interactions increases as allowed by changes in thermodynamics (temperature drop) and biophysical constraints. This process fosters also the emergence of scale. Indeed, while an isolated, ideal thermodynamic system (like a gas) is usually a scale-free system, the establishment of relevant cooperativity among molecules leads to the emergence of scale-dependent observables. As the number of correlations increases among the components, nonlinear dynamic interactions emerge. Then, within the mesoscopic realm, we witness the constitution of organized and dynamical complexity, as documented by the development of networks among the components of the system. In turn, networks promote the development of structures, the establishment of internal/external boundaries (a critical step epitomized by the appearance of membranes), which overall contribute in the emergence of biophysical and structural constraints. The system becomes “compartmentalized” and the consequent birth of intracellular gradients and interfaces adds further complexity in modulating the system’s dynamics and interconnectivity. The potential degrees of freedom are progressively “reduced” as the organization in the system increases. Now, constraints and control factors (both internal and external) play a critical role in shaping the overall architecture and functioning of the system, which cannot be described by classical thermodynamics anymore. The system becomes “dissipative.” As such, it can be understood by advocating the non-equilibrium theory. Complex entities are dynamical, as they modify their entropy and structure over time, leading to the appearance of emergent properties that cannot be anticipated by studying the components as isolated from the system as a whole. Notice that, at the top, we allocated a diamond as an example of full-organized structure without significant complexity, as it does not show changes over time. Therefore, the key feature is represented by changes occurring over time among the dynamical relationships and their constraints (while preserving their ordered structure). This interplay is likely to explain the remarkable characteristics of complex systems.

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
