# Peer review of "Complexity in Biological Organization: Deconstruction (and Subsequent Restating) of Key Concepts"

_entropy, 2020, doi:10.3390/e22080885_

Round 1

Reviewer 1 Report

The review paper by Bizzarri and collaborators presents an interesting and timely account of biological complexity based on both a historical account and a constructivist 'bottom up" introduction. The authors are well-known scholars in the physics of biological complexity so the work is authoritative.

My only minor comment will be in the isue of presentation. The review is written in the form of an assay, which is a form preferred by the experts on a certain topic to discuss among themselves. If this is their intention, then the work is flawless. If, on the other hand, an introductory presentation to non-experts is desired, then I will recommend to supplement their work with diagrams, or workflows, even tables to serve as roadmap to guide the reader into the discussion.

A few minor "typos" can be found in the manuscript (e.g., "dee" instead of "deep" in the abstract) that the authors may want to correct.

Reviewer 2 Report

The authors discussed what "complexity" should include when studying biological systems (and non-living systems). The authors showed profound knowledge in theories and methodologies in systems biology studies and comprehensively knitted their points of views in one paper. My research experience only covers a small proportion of those mentioned in the manuscript so I'll just comment based on my limited expertise.

Major comments:
1. The authors decomposed the term "complexity" into different aspects. The information carried in the text are massive and seems comprehensive. I believe that the authors have a well-knitted knowledge network in mind, but since there's a huge amount of information, the readers can get overwhelmed and it may be difficult to find the interconnections between the concepts described in the paper. Could the authors consider adding a "knowledge structure map" to demonstrate the different compartments?

2. In some part of the manuscript, since I have some experience I could map the contents to my knowledge. For example, when the authors mention relationship, constraints and dynamics I thought about equations of mathematical physics and dynamic system modeling. The authors also mentioned a few concepts in network science that I can quickly map to. However there are also my knowledge gaps and imagine readers like me, would expect the authors mention the name of subject that a concept is from, and what methodology from this subject should be used to solve these questions.

3. The different examples that the authors used are too broad, covering living and non-living systems, from water, to DNA and cell, and our Universe... Could the authors specify the focus on a specific type fo biological system?

4. Then it comes to the most interesting part, and which is my biggest concern: the authors describe different levels of biological organization, different components of complexity, and many examples to explain all the different concepts. A big missing link is how are we suppose to solve the problem of system complexity step-by-step? Is there a suggested way to systematically solve a biological system, unveiling the interconnections between different layers, using a comprehensive interdisciplinary knowledge? I feel that this should be the essence of this paper but it's not found in the current version. Now it's more like a hodgepodge and readers can easily get lost.

Round 2

Reviewer 2 Report

I appreciate the authors' efforts in addressing the comments.

There are two further comments:
1) Could the authors add some detailed phrases on the figure unde each node?
2) Usually pictures from online sources are not accepted because of copyright. Please make sure picture used are orginally created by the authors.
